# Specific PCR primer designed from genome data for rapid detection of *Fusarium oxysporum* f. sp. *cubense* tropical race 4 in the Cavendish banana

**Shunsuke Nozawa[1], Dan Charlie Joy Pangilinan[1], G. Alvindia Dionisio[2], Kyoko Watanabe[1] ***

**1** School of Agriculture, Tamagawa University, Machida, Tokyo, Japan, **2** Philippine Center for Postharvest Development and Mechanization, Nueva Ecija, Philippines

* wkyoko@agr.tamagawa.ac.jp

**Data Availability Statement:** All relevant data are within the manuscript and its Supporting information files.

## Abstract

Fusarium wilt caused by *Fusarium oxysporum* f. sp. *cubense* (Foc) tropical race 4 (TR4) severely affects banana production worldwide. Thus, specific PCR primers have been developed to rapidly diagnose and monitor Foc TR4-related fusarium wilt outbreaks in bananas. However, evaluation of these primers revealed room for improvement in the accuracy. This study aimed to design highly specific PCR primers based on genome data for Foc TR4 downloaded from the National Center for Biotechnology Information database. The specificity of the primers was assessed using Foc TR4, Foc races 1 and 2, and 15 other formae speciales strains. The utility of the primers was verified by correctly detecting Foc TR4 in 7 out of 86 isolates of *Fusarium* spp. obtained from banana farms in the Philippines. The primers allowed for rapid detection in experimentally diseased tissues. We concluded that this novel primer set enables the simplified diagnosis of fusarium wilt caused by Foc TR4 in bananas.

## Introduction

Bananas are an important cash crop in Southeast Asia, Africa, and Latin America. According to FAOSTAT, banana production reached 1.25 million tons in 2021. However, banana production has been threatened by banana wilt disease caused by *Fusarium oxysporum* f. sp. *cubense* (Foc), a soil-borne pathogen. In the 20th century, the disease caused by Foc race 1 eradicated thousands of hectares of the Gros Michel banana in Central America. In response to this, the resistant variety, Cavendish banana was discovered and this variety became the dominant commercial banana worldwide. However, since its 1989 outbreak in a Cavendish plantation in Taiwan [1, 2], fusarium wilt caused by Foc tropical race 4 TR4 (syn. *Fusarium odoratissimum*) has severely affected banana production worldwide. Since Foc TR4 produces durable chlamydospores that can survive in the soil for more than a decade without a host,

**Funding:** JICA JST SATREPS Grant Number
JPMJSA2007.

during which susceptible varieties cannot be successfully replanted [3, 4]. Given their importance in food production, worldwide monitoring of disease outbreaks has been implemented.

Foc TR4 has been identified through pathogenicity testing of susceptible varieties and determining vegetative compatibility groups (VCGs) using nitrate non-utilizing (nit) mutants [5]. However, pathogenicity and VCG testing may take 1–2 months, and VCGs cannot be identified without reference strains. A recent study showed that although it is difficult to find differences in morphology from other races, Foc TR4 is independent of other lineages including Foc race 1 and Foc subtropical race 4 (STR4), based on phylogenetic analysis using four loci: calmodulin (*cmdA*), RNA polymerase second largest subunit (*rpb2*), translation elongation factor 1-alpha (*tef1*), and β-tubulin 2 (*tub2*) regions [6]. Therefore, Foc TR4 can be distinguished from other races by phylogenetic analysis. However, this method also requires >1 month for isolation, purification, DNA extraction, polymerase chain reaction (PCR), sequencing, and phylogenetic analysis. Thus, conventional detection methods are unsuitable for the rapid diagnosis or investigation of disease outbreaks.

To address the current gap in the field, Foc TR4-specific PCR primers have been designed. Dita et al. [7] and Li et al. [8] accomplished this using the intergenic spacer region of the rDNA cluster (IGS), while Li et al. [9] and Carvalhais et al. [10] designed primers based on pathogen-related genes encoded by xylem 1 and W2987, respectively. However, because conserved regions, such as the IGS, cannot accurately distinguish differences [11], false-positive rates are high [12]. Using the primers designed for pathogen-related genes by Li et al. [9] and Carvalhais et al. [10], Yang et al. [12] revealed that some Foc TR4 strains were not detected and disease incidence may be underestimated. In this study, we aimed to develop a specific primer set based on the genome data for the Foc TR4-specific site. This primer set represents a promising epidemiological tool for the rapid diagnosis of fusarium wilt.

## Materials and methods

### Phylogenetic analysis using genome data

We performed a genome-scale phylogenetic analysis based on a concatenation approach to confirm that the Foc TR4 strains were monophyletic and determine the phylogenetic relationships between Foc TR4 and other races and formae speciales. The genome data of 32 formae speciales (93 strains, including Foc TR4) were retrieved from the National Center for Biotechnology Information (NCBI) database (S1 Table) and used as an ingroup. *F. fujikuroi* (IMI58289) was retrieved from the NCBI database and used as an outgroup. Gene predictions were performed with Augustus v.3.3.3 [13] with the following parameter: "—species = fusarium <genomic data>". Orthology inference was performed by reciprocal BLAST analysis (BLASTP) with BLAST 2.9.0+ software ($e \leq$ 1e-05) [14, 15]. Multiple alignments for each orthologous gene were conducted using Clustal Omega v.1.2.2 [16] with the default settings at the translated amino acid sequence level. To select genes comprising >1,000 bp after alignment, we trimmed gap-including sites with trimAl v.1.2 [17] and counted the remaining sites using PhyKIT v.1.2.0 (https://jlsteenwyk.com/PhyKIT/ [18]). Phylogenetic trees were constructed using a concatenation approach. The concatenated alignment of DNA data from all gene sets (4,595 genes) was analyzed using RAxML v.7.0.4 [19] under the GTRGAMMA model with 100 bootstrap replicates. A phylogenetic tree was drawn using iTOL v.6 (https://itol.embl.de/).

### Primer design

We used the genome data for 21 strains of Foc TR4 and 93 strains of 32 different formae speciales (the same dataset used to construct the genome-scale phylogenetic trees) to explore Foc

TR4-specific genes. Gene prediction and orthology inferences were performed as described in **Phylogenetic analysis using genome data**. Genes found only in Foc TR4, excluding BC2-4, were selected as candidates. BLAST analyses were conducted using the DNA sequence data of candidate genes against the genome data to confirm their detection in genome regions where they were not previously predicted. Primers were designed using Primer3Plus (https://www.primer3plus.com/) with the default parameters for melting temperature, GC content, and primer length. The presence of the sequence was verified through a BLAST search on the NCBI website (https://blast.ncbi.nlm.nih.gov/Blast.cgi?PROGRAM=blastn&PAGE_TYPE= BlastSearch&LINK_LOC=blasthome).

## Assessment of primer set specificity

To compare the accuracy between our primer set and those designed by Carvalhais et al. [10], Dita et al. [7], Li et al. [9], and Li et al. [8], the primers were also tested under reported PCR conditions (S2 Table). Foc TR4 isolates (2718M; S1 Fig) that confirmed its pathogenicity against the Cavendish variety and (S2 Fig), Foc race1, Foc race2, and 15 other formae speciales (25 isolates, Table 1) deposited at the National Agricultural and Food Research Organization (NARO) Genetic Resources Research Centre (Kumamoto, Japan) were used for PCR. Template DNA for PCR was obtained from the mycelia of each isolate grown on potato dextrose agar for 7–10 d using a modified CTAB method [20], as described by Nozawa et al. [21]. PCR was performed using a 10 μL PCR mixture containing 7 μL distilled water, 1 μL 10× Ex Taq buffer with $MgCl_2$, 0.8 μL dNTP (10 mM each), 0.1 μL of each primer (50 μM), 0.05 μL Ex Taq DNA polymerase (5 U $μL^{-1}$, Takara, Tokyo, Japan), and 1.0 μL of DNA template. Amplification of the partial 18S rRNA gene, ITS1, 5.8S rRNA gene, ITS2, and partial 28S rRNA gene [internal transcribed spacer (ITS) region] was performed to confirm successful DNA extraction. The PCR conditions described by White et al. [22] were used to amplify the ITS region. The thermal cycling program for primers 13721F and 13712R consisted of an initial denaturing step of 5 min at 95 ˚C, followed by 35 cycles of 10 s at 95 ˚C, 5 s at 56 ˚C, and 10 s at 72 ˚C, with a final extension step of 3 min at 72 ˚C.

## Investigating Foc TR4 in isolates from the Philippines

Symptomatic plants were collected from banana farms in Mindanao in 2022 to obtain banana isolates. Discolored vascular tissues of the pseudostem were cut into 5 × 5 mm pieces that were then sterilized with 0.6% (v/v) sodium hypochlorite for 1 min, washed with sterilized water, dried with sterilized paper, and placed on a water–agar (WA) plate. Hyphae that emerged on the WA were transferred onto potato dextrose agar plates to produce conidia for monoculture. The obtained 86 isolates of *Fusarium* spp. were used for subsequent experiments.

PCR amplification of the 86 isolates was conducted using primer sets 13712F and 13712R. A 200-bp amplicon was confirmed by gel electrophoresis. We confirmed that the detected isolates were clustered with Foc TR4. First, to select isolates belonging to the *F. oxysporum* species complex Foc TR4, molecular phylogenetic analysis was performed on the 86 isolates and ex-type strains of *Fusarium* species based on the ITS region (S1 Dataset).

We performed a molecular phylogenetic analysis of the 21 isolates clustered with the *F. oxysporum* species complex based on the concatenated data for *cmdA*, *rpb2*, *tef1*, and *tub2* regions as previously described [6] to reveal the phylogenetic positions of the isolates (S2 Dataset). We selected 20 species, consisting of 54 strains in the *F. oxysporum* species complex, reported by Lombard et al. [6]; *F. foetens* (CBS120665) and *F. udum* (CBS130302) were used as outgroups (Table 2). Alignment of each sequence dataset was performed using ClustalW in BioEdit v.7.2 [23]. The aligned datasets were combined for phylogenetic analyses using neighbor-joining

**Table 1. Strains of pathogenic *F. oxysporum* and results of PCR using primers for specific detection of FocTR4.**

| Strain | f. sp. (race) | Host | Primer sets | | | | |
|---|---|---|---|---|---|---|---|
| | | | 13712F 13712R (this study) | FocTR4F FocTR4R (Dita et al. 2010) | W2987F W2987R (Li et al., 2013b) | SIX1a-266-F SIX1a-266-R (Carvalhais et al., 2019) | VCG01213 16F1 VCG01213 16R2 (Li et al., 2013a) |
| 2718M | cubense (TR4) | banana | + | + | - | + | + |
| race1 | cubence (race 1) | banana | - | - | - | - | + |
| race2 | cubence (race 2) | banana | - | + | - | - | + |
| MAFF 103008 | lagenariae | bottle gourd | - | + | - | - | - |
| MAFF 103036 | lycopersici | tomato | - | - | - | - | + |
| MAFF 103051 | melogenae | eggplant | - | + | - | - | + |
| MAFF 103054 | cucumerinam | cucumber | - | + | - | - | + |
| MAFF 103059 | spinaciae | spinach | - | - | - | - | + |
| MAFF 235105 | tulipae | tulip | - | - | - | - | - |
| MAFF 235154 | not determine | rice | - | + | - | - | + |
| MAFF 237022 | not determine | taro | - | + | - | - | - |
| MAFF 240102 | tanaceti | feverfew | - | + | - | - | + |
| MAFF 240327 | rapae | tatsoi | - | - | - | - | - |
| MAFF 240804 | not determine | bitter gourd | - | + | - | - | + |
| MAFF 240805 | not determine | bitter gourd | - | + | - | - | + |
| MAFF 241054 | adzukicala | azuki bean | - | + | - | - | + |
| MAFF 243255 | lactucae | lettuce | - | + | - | - | + |
| MAFF 243476 | not determine | delphinium | - | + | - | - | + |
| MAFF 305543 | niveum | watermelon | - | - | - | - | + |
| MAFF 305544 | melonis | melon | - | + | - | - | + |
| MAFF 305606 | batatas | sweet potato | - | + | + | + | - |
| MAFF 305608 | niveum | watermelon | - | - | - | - | + |
| MAFF 305937 | radicis-lycopersici | tomato | - | + | - | - | + |
| MAFF 306313 | not determine | taro | - | + | - | - | + |
| MAFF 727508 | cacumerinum | cucumber | - | + | - | - | + |
| MAFF 744004 | cacumerinum | cucumber | - | + | - | - | + |
| MAFF 744088 | lactucae | lettuce | - | - | - | - | + |
| MAFF 306716 | cubence | banana | - | + | - | - | + |

'+' indicates that the relevant strain was detected with the primers.

'-' indicated that the relevant strain was not detected with the primers.

(NJ), maximum-likelihood (ML), and maximum-parsimony (MP) algorithms in MEGA10 [24]. Gap-including sites were treated as missing data. The reliability of the internal branches in the trees was evaluated through a 1,000 replicate bootstrap analysis [25].

## Detection in plant tissues

PCR amplification was conducted using DNA extracted from banana tissues to demonstrate the diagnosis of diseased tissues. A pathogenicity test was conducted to obtain diseased banana tissues as previously described [21]. Tissues (0.2 g) from corms with dark brown to black lesions and healthy tissues (without inoculation) were used for DNA extraction using the CTAB method. PCR amplification was conducted three times under the conditions described in **Assessment of primer set specificity**. DNA templates extracted from healthy bananas and sterile water were used as negative controls, and mycelia of Foc TR4 were used as positive controls. DNA extraction was confirmed by gel electrophoresis.

**Table 2. Strains and Genebank accession number using phylogenetic analysis.**

| Species | Strain no. | Accession no. | | | |
|---|---|---|---|---|---|
| | | *cmdA* | *rpb2* | *tef1* | *tub2* |
| *F. callistephi* | CBS187.53 | MH484693 | MH484875 | MH484966 | MH485057 |
| | CBS115423 | MH484723 | MH484905 | MH484996 | MH485087 |
| *F. carminascens* | CBS144739 | MH484752 | MH484934 | MH485025 | MH485116 |
| | CBS144740 | MH484753 | MH484935 | MH485026 | MH485117 |
| | CBS144741 | MH484754 | MH484936 | MH485027 | MH485118 |
| *F. contaminatum* | CBS111552 | MH484718 | MH484900 | MH484991 | MH485082 |
| | CBS114899 | MH484719 | MH484901 | MH484992 | MH485083 |
| | CBS117461 | MH484729 | MH484911 | MH485002 | MH485093 |
| *F. cugenangense* | CBS620.72 | MH484697 | MH484879 | MH484970 | MH485061 |
| | CBS130304 | MH484739 | MH484921 | MH485012 | MH485103 |
| | CBS130308 | MH484738 | MH484920 | MH485011 | MH485102 |
| *F. curvatum* | CBS247.61 | MH484694 | MH484876 | MH484967 | MH485058 |
| | CBS238.94 | MH484711 | MH484893 | MH484984 | MH485075 |
| | CBS141.95 | MH484712 | MH484894 | MH484985 | MH485076 |
| *F. duoseptatum* | CBS102026 | MH484714 | MH484896 | MH484987 | MH485078 |
| *F. elaeidis* | CBS217.49 | MH484688 | MH484870 | MH484961 | MH485052 |
| | CBS218.49 | MH484689 | MH484871 | MH484962 | MH485053 |
| | CBS255.52 | MH484692 | MH484874 | MH484965 | MH485056 |
| *F. fabacearum* | CBS144742 | MH484756 | MH484938 | MH485029 | MH485120 |
| | CBS144743 | MH484757 | MH484939 | MH485030 | MH485121 |
| | CBS144744 | MH484758 | MH484940 | MH485031 | MH485122 |
| *F. foetens* | CBS120665 | MH484736 | MH484918 | MH485009 | MH485100 |
| *F. glycines* | CBS176.33 | MH484686 | MH484868 | MH484959 | MH485050 |
| | CBS214.49 | MH484687 | MH484869 | MH484960 | MH485051 |
| | CBS200.89 | MH484706 | MH484888 | MH484979 | MH485070 |
| *F. gossypinum* | CBS116611 | MH484725 | MH484907 | MH484998 | MH485089 |
| | CBS116612 | MH484726 | MH484908 | MH484999 | MH485090 |
| | CBS116613 | MH484727 | MH484909 | MH485000 | MH485091 |
| *F. hoodiae* | CBS132474 | MH484747 | MH484929 | MH485020 | MH485111 |
| | CBS132476 | MH484748 | MH484930 | MH485021 | MH485112 |
| | CBS132477 | MH484749 | MH484931 | MH485022 | MH485113 |
| *F. languescens* | CBS645.78 | MH484698 | MH484880 | MH484971 | MH485062 |
| | CBS646.78 | MH484699 | MH484881 | MH484972 | MH485063 |
| | CBS413.90 | MH484708 | MH484890 | MH484981 | MH485072 |
| *F. libertatis* | CBS144748 | MH484750 | MH484932 | MH485023 | MH485114 |
| | CBS144747 | MH484751 | MH484933 | MH485024 | MH485115 |
| | CBS144749 | MH484762 | MH484944 | MH485035 | MH485126 |
| *F. nirenbergiae* | CBS129.24 | MH484682 | MH484864 | MH484955 | MH485046 |
| | CBS149.25 | MH484683 | MH484865 | MH484956 | MH485047 |
| | CBS840.88 | MH484705 | MH484887 | MH484978 | MH485069 |
| *F. odoratissimum* | CBS794.70 | MH484696 | MH484878 | MH484969 | MH485060 |
| | CBS102030 | MH484716 | MH484898 | MH484989 | MH485080 |
| | CBS102030 | MH484740 | MH484922 | MH485013 | MH485104 |
| *F. oxysporum* | CBS130310 | MH484690 | MH484872 | MH484963 | MH485054 |
| | CBS221.49 | MH484771 | MH484953 | MH485044 | MH485135 |
| | CBS144134 | MH484772 | MH484954 | MH485045 | MH485136 |

(*Continued*)

**Table 2.** (Continued)

| Species | Strain no. | Accession no. | | | |
|---|---|---|---|---|---|
| | | *cmdA* | *rpb2* | *tef1* | *tub2* |
| *F. pharetrum* | CPC25822 | MH484769 | MH484951 | MH485042 | MH485133 |
| | CBS144750 | MH484770 | MH484952 | MH485043 | MH485134 |
| *F. trachichlamydosporum* | CBS144751 | MH484715 | MH484897 | MH484988 | MH485079 |
| *F. triseptatum* | CBS102028 | MH484691 | MH484873 | MH484964 | MH485055 |
| | CBS258.50 | MH484728 | MH484910 | MH485001 | MH485092 |
| | CBS116619 | MH484734 | MH484916 | MH485007 | MH485098 |
| *F. udum* | CBS130302 | MH484684 | MH484866 | MH484957 | MH485048 |
| *F. veterinarium* | CBS177.31 | MH484717 | MH484899 | MH484990 | MH485081 |
| | CBS109898 | MH484730 | MH484912 | MH485003 | MH485094 |
| | CBS117787 | MH484731 | MH484913 | MH485004 | MH485095 |

## Results

### Phylogenetic relationship between Foc TR4 and other races as well as formae speciales

To confirm that Foc TR4 is monophyletic, we constructed a molecular phylogenetic tree based on the genome data deposited on NCBI. We identified 4,595 orthologous genes through reciprocal BLAST searches. The alignment length, number of variable sites, and number of parsimony-informative sites in the DNA data were 8,356,161 bp, 923,458, and 345,950, respectively. The ML tree showed that the 21 Foc TR4 strains were monophyletic, supported with a 100% bootstrap value (Fig 1).

### Primer design

Among the Foc TR4-specific genes from the genome data, gene 13712 was found to be best for primer design, though it was absent in strain BC2-4 (Foc TR4) and present in strains Foc011 and Foc013 (f. sp. *cucumerinum*, Fig 1). Strains Foc011 and Foc013 with gene 13712 were monophyletic (100% bootstrap value), and polyphyletic with the Foc TR4 clade. These two strains separated into a lineage consisting of f. sp. *cubense* subtropical race 4 (STR4, VCG0120) and f. sp. *cubense* (VPRI44083, race not determined).

The primer set 13712F (5′-CTG AGGA TAG CAC TTG TTT T-3′) and 13712R (5′-AAA GAC TAT AGG TAT GCT TTA ATC A-3′), which produces a 201-bp PCR amplicon (S1 Data), was designed based on the DNA sequence of gene 13712. For a forward primer 13712F, GC content is 40%, and the annealing temperature is 53.2˚C. For a reverse primer13712R, GC content is 26.9%, and the annealing temperature is 54.6˚C. The primer sites were completely conserved among the 20 Foc TR4 strains, except for the BC2-4 strain (Fig 2).

### Assessment of primers 13712F and 13712R

We performed PCR amplification of a 200-bp DNA fragment from the Foc TR4 strain using primers 13712F and 13712R (Table 1, S3 Fig). This is consistent with the expected length from the genome data. To confirm the identity of the amplified fragment, the sequence was compared to the target region obtained from the genome data. The resulting fragment sequence was 100% identical to the corresponding sequence in the genome data. No PCR DNA amplicons were amplified from other *F. oxysporum* formae speciales, including *adzukicala, batatas, cubense, cucumerinam, lactucae, lagenariae, lycopersici, melogenae, melonis, niveum, radicis-*

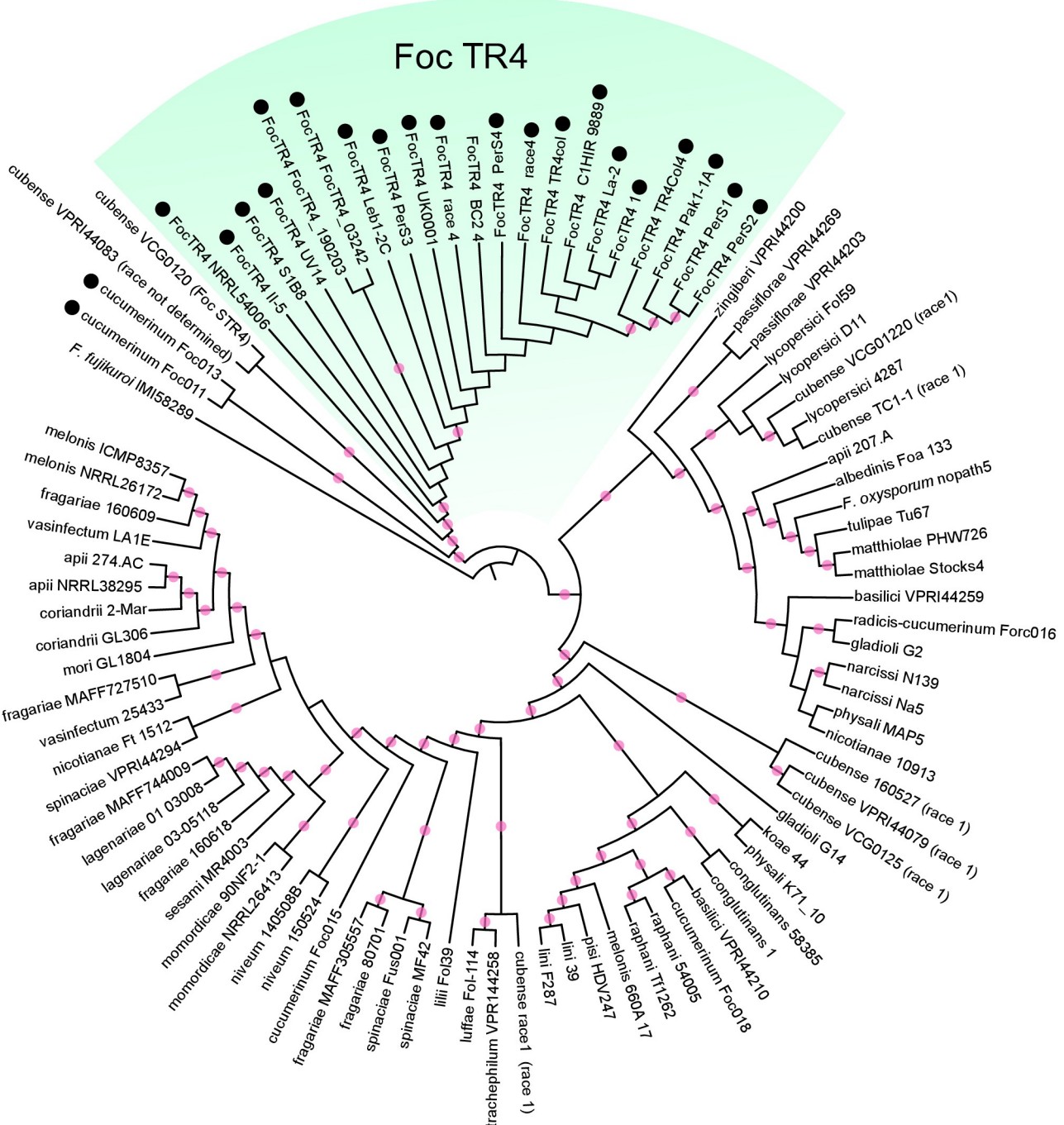

**Fig 1. Phylogenetic trees based on the concatenated DNA sequence data for 4,595 orthologous genes of the protein-coding region.** Red circles on internal branches indicate 90% bootstrap (BS) values. Isolates with a black circle carry gene 13712, from which primers were designed. The green-highlighted clade is a Foc TR4 clade.

*lycopersici, rapae, spinaciae, tanaceti,* and *tulipae,* or strains with no formae speciales assignment but showing pathogenicity against bitter gourd, delphinium, rice, and taro. In contrast, the primers designed previously for the detection of Foc TR4 by Dita et al. [7], Carvalhais et al. [10], and Li et al. [9] amplified 18, 1, and 21 strains other than Foc TR4. The primers designed

|  | 13712F | 13712R |
|---|---|---|
|  | 1                    20 | 185                    200 |
| FocTR4_032 | CTGAGGATAG CACTTGTTTT AGTAGAAGAG | ATTTTTGATT AAAGCATACC TATAGTCTTT |
| FocTR4_190 | CTGAGGATAG CACTTGTTTT AGTAGAAGAG | ATTTTTGATT AAAGCATACC TATAGTCTTT |
| La-2 | CTGAGGATAG CACTTGTTTT AGTAGAAGAG | ATTTTTGATT AAAGCATACC TATAGTCTTT |
| Leb1-2C | CTGAGGATAG CACTTGTTTT AGTAGAAGAG | ATTTTTGATT AAAGCATACC TATAGTCTTT |
| Pak1-1A | CTGAGGATAG CACTTGTTTT AGTAGAAGAG | ATTTTTGATT AAAGCATACC TATAGTCTTT |
| JV14 | CTGAGGATAG CACTTGTTTT AGTAGAAGAG | ATTTTTGATT AAAGCATACC TATAGTCTTT |
| 1 | CTGAGGATAG CACTTGTTTT AGTAGAAGAG | ATTTTTGATT AAAGCATACC TATAGTCTTT |
| TR4Col | CTGAGGATAG CACTTGTTTT AGTAGAAGAG | ATTTTTGATT AAAGCATACC TATAGTCTTT |
| TR4Col4 | CTGAGGATAG CACTTGTTTT AGTAGAAGAG | ATTTTTGATT AAAGCATACC TATAGTCTTT |
| II-5 | CTGAGGATAG CACTTGTTTT AGTAGAAGAG | ATTTTTGATT AAAGCATACC TATAGTCTTT |
| S1B8 | CTGAGGATAG CACTTGTTTT AGTAGAAGAG | ATTTTTGATT AAAGCATACC TATAGTCTTT |
| PerS1 | CTGAGGATAG CACTTGTTTT AGTAGAAGAG | ATTTTTGATT AAAGCATACC TATAGTCTTT |
| PerS2 | CTGAGGATAG CACTTGTTTT AGTAGAAGAG | ATTTTTGATT AAAGCATACC TATAGTCTTT |
| PerS3 | CTGAGGATAG CACTTGTTTT AGTAGAAGAG | ATTTTTGATT AAAGCATACC TATAGTCTTT |
| PerS4 | CTGAGGATAG CACTTGTTTT AGTAGAAGAG | ATTTTTGATT AAAGCATACC TATAGTCTTT |
| C1HIR_9889 | CTGAGGATAG CACTTGTTTT AGTAGAAGAG | ATTTTTGATT AAAGCATACC TATAGTCTTT |
| Foc4_1.0 | CTGAGGATAG CACTTGTTTT AGTAGAAGAG | ATTTTTGATT AAAGCATACC TATAGTCTTT |
| UK0001 | CTGAGGATAG CACTTGTTTT AGTAGAAGAG | ATTTTTGATT AAAGCATACC TATAGTCTTT |
| NRRL54006 | CTGAGGATAG CACTTGTTTT AGTAGAAGAG | ATTTTTGATT AAAGCATACC TATAGTCTTT |
| race_4 | CTGAGGATAG CACTTGTTTT AGTAGAAGAG | ATTTTTGATT AAAGCATACC TATAGTCTTT |

**Fig 2. Alignment of primer sitting on gene 13712 of Foc TR4 strains.** Highlighted sites are primer sites.

by Li et al. [8] did not amplify the 452-bp target region of Foc TR4. Therefore, the primer set designed in this study showed more specificity for Foc TR4 than previously reported primers.

## Detection of Foc TR4 from banana farms

Based on the ITS region, 21 of the 86 isolates collected from banana farms in the Philippines were clustered with the *F. oxysporum* species complex (92% bootstrap value) in the phylogenetic tree (Fig 3). Molecular phylogenetic analysis of the concatenated data for the four loci (2,378 bp), *cmdA* (423 bp), *rpb2* (877 bp), *tef1* (620 bp), and *tub2* (458 bp) identified seven isolates forming a monophyletic clade with *F. odoratissimum* strains, which are considered to be Foc TR4 (ML/NJ/MP = 94%/97%/99% bootstrap value; Fig 4). These isolates were effectively detected through PCR using the primer set 13712F and 13712R (Table 3). Other isolates were clustered with *F. elaeidis* (5 isolates; ML/NJ/MP = 73%/71%/84%), *F. fabacearum* (one isolate; ML/NJ/MP = -/-/78%), *F. trachichlamydosporum* (two isolates; ML/NJ/MP = 98%/98%/99%), and *F. triseptatum* (two isolates; ML/NJ/MP = 99%/99%/99%). Four isolates are independent of known species and closely related to *F. glycines*. One isolate, PH22-1069, of those, and three isolates, PH22-1006, PH22-1011, and PH22-1023, from other species complexes (not determined; Fig 3) were detected using this primer set were not Foc TR4 (Figs 3 and 4, S4 Fig).

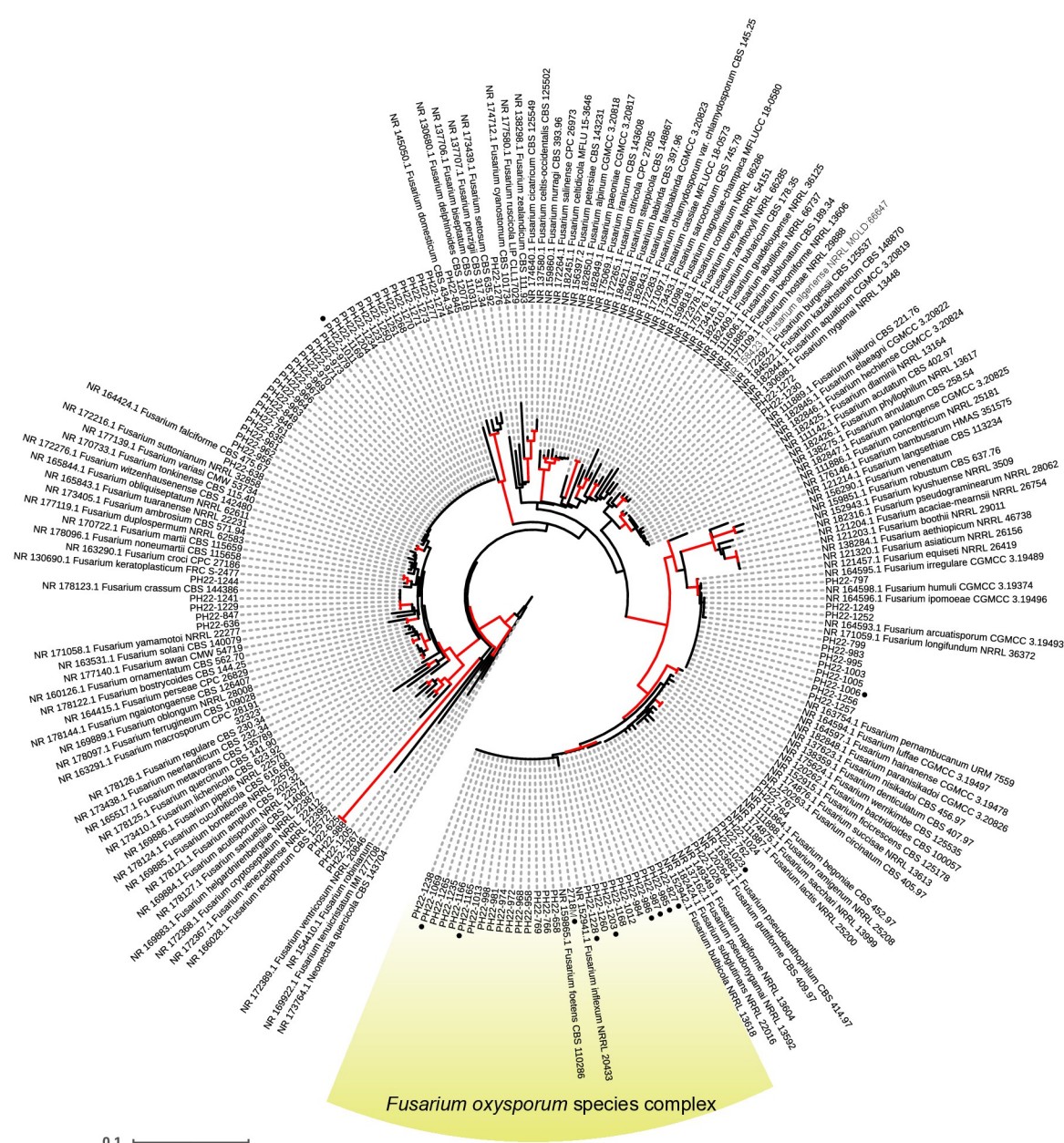

**Fig 3. Phylogenetic relationships of isolates in this study and known species inferred using the sequence for the ITS region.**
Evolutionary history was inferred using the NJ method. The tree is drawn to scale, with branch lengths in the same units as those of the evolutionary distances used to infer the phylogenetic tree. The evolutionary distances were computed using the p-distance method and are expressed as the number of base differences per site. There were a total of 472 positions in the final dataset. Evolutionary analyses were conducted in MEGA10. Isolates with black dots were detected using the primer set designed in this study. Red branches have >70% bootstrap values.

## Detection in tissues

The 200-bp PCR amplicons were only obtained from diseased tissues experimentally infected with Foc TR4 and Foc TR4 mycelia (Fig 5). No PCR amplicons were obtained from healthy tissues or water.

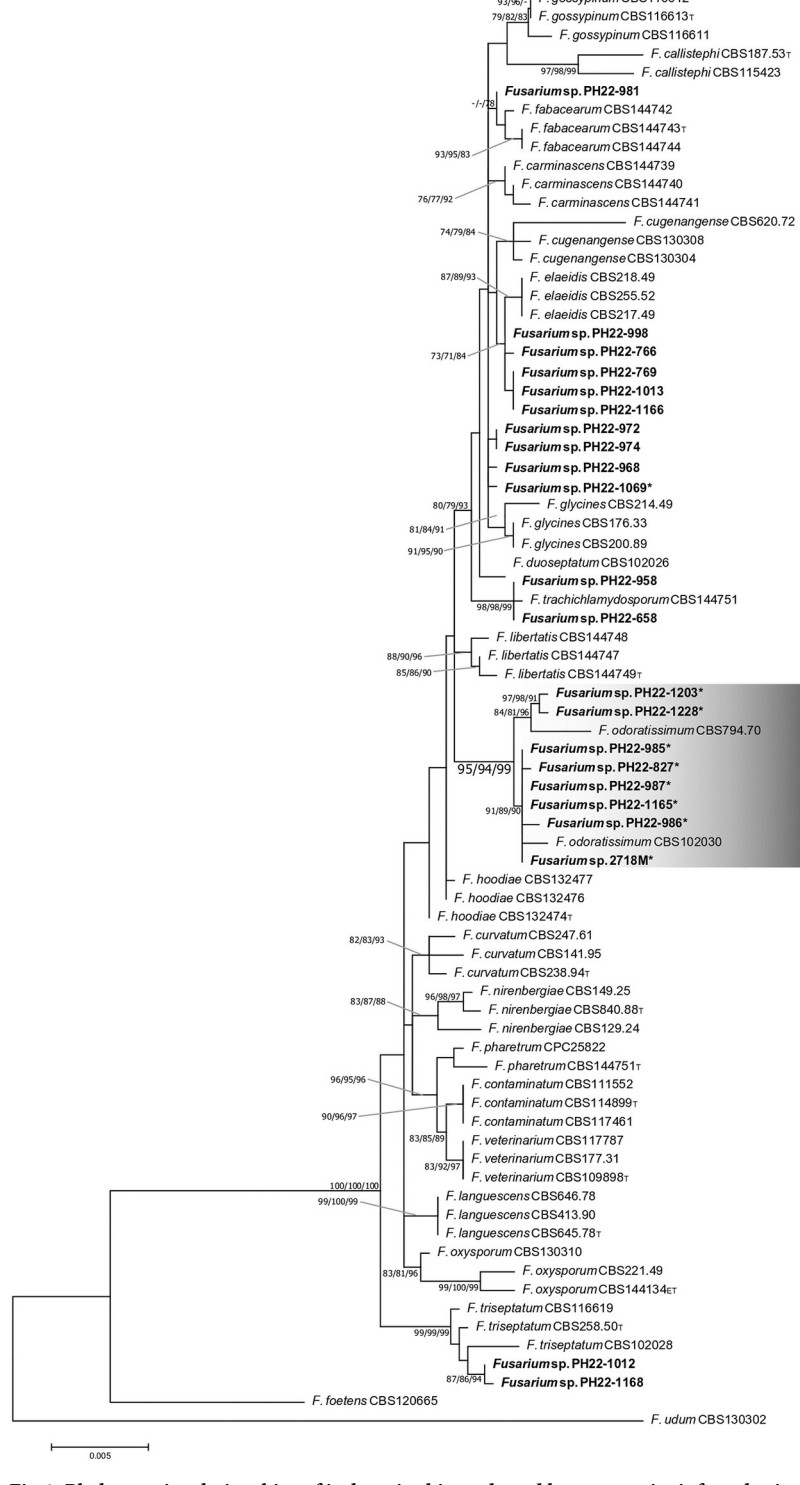

**Fig 4. Phylogenetic relationships of isolates in this study and known species inferred using the concatenated data for the four loci (2,378 bp), *cmdA* (423 bp), *rpb2* (877 bp), *tef1* (620 bp), and *tub2* (458 bp).** Evolutionary history was inferred using the ML method. The values accompanying internal branches are bootstrap values (ML/NJ/MP, ≥70%). -, absence of node; T, ex-type strain; ET, ex-epitype strain. Isolates with black dots were detected using the primer set designed in this study.

**Table 3. Results of PCR detection using primer set 13712F/13712R from isolates in the Philippines, their phylogenetic positions and location.**

| Isolates | Detection | Phylogenetic position | Location |
|---|---|---|---|
| PH22-0625 | - | *Fusarium* sp. (not *F. oxysporum* species complex) | Balila, Lantapan, Bukidnon |
| PH22-0635 | - | *Fusarium* sp. (not *F. oxysporum* species complex) | Midland, Valencia City, Bukidnon |
| PH22-0636 | - | *Fusarium* sp. (not *F. oxysporum* species complex) | Midland, Valencia City, Bukidnon |
| PH22-0638 | - | *Fusarium* sp. (not *F. oxysporum* species complex) | Midland, Valencia City, Bukidnon |
| PH22-0658 | - | cf. *F. trachichlamydosporum* | Valencia City, Bukidnon |
| PH22-0761 | - | *Fusarium* sp. (not *F. oxysporum* species complex) | Tibal-og (Poblacion), Sto. Tomas, Davao del Norte |
| PH22-0762 | - | *Fusarium* sp. (not *F. oxysporum* species complex) | Tibal-og (Poblacion), Sto. Tomas, Davao del Norte |
| PH22-0764 | - | *Fusarium* sp. (not *F. oxysporum* species complex) | Tibal-og (Poblacion), Sto. Tomas, Davao del Norte |
| PH22-0766 | - | cf. *F. elaeidis* | Tibal-og (Poblacion), Sto. Tomas, Davao del Norte |
| PH22-0767 | - | *Fusarium* sp. (not *F. oxysporum* species complex) | Tibal-og (Poblacion), Sto. Tomas, Davao del Norte |
| PH22-0769 | - | cf. *F. elaeidis* | Tibal-og (Poblacion), Sto. Tomas, Davao del Norte |
| PH22-0797 | - | *Fusarium* sp. (not *F. oxysporum* species complex) | Sto. Tomas, Davao del Norte |
| PH22-0799 | - | *Fusarium* sp. (not *F. oxysporum* species complex) | Sto. Tomas, Davao del Norte |
| PH22-0827 | + | Foc TR4 (cf. *F. odoratissimum*) | Davao Coastal Road, Mabini, Davao De Oro |
| PH22-0845 | - | *Fusarium* sp. (not *F. oxysporum* species complex) | Davao Coastal Road, Mabini, Davao De Oro |
| PH22-0846 | - | *Fusarium* sp. (not *F. oxysporum* species complex) | Davao Coastal Road, Mabini, Davao De Oro |
| PH22-0847 | - | *Fusarium* sp. (not *F. oxysporum* species complex) | Davao Coastal Road, Mabini, Davao De Oro |
| PH22-0849 | - | *Fusarium* sp. (not *F. oxysporum* species complex) | Davao Coastal Road, Mabini, Davao De Oro |
| PH22-0956 | - | *Fusarium* sp. (not *F. oxysporum* species complex) | Finca Verde, Tugbok, Davao City, Davao del Sur |
| PH22-0958 | - | cf. *F. trachichlamydosporum* | Finca Verde, Tugbok, Davao City, Davao del Sur |
| PH22-0961 | - | *Fusarium* sp. (not *F. oxysporum* species complex) | Finca Verde, Tugbok, Davao City, Davao del Sur |
| PH22-0962 | - | *Fusarium* sp. (not *F. oxysporum* species complex) | Finca Verde, Tugbok, Davao City, Davao del Sur |
| PH22-0963 | - | *Fusarium* sp. (not *F. oxysporum* species complex) | Finca Verde, Tugbok, Davao City, Davao del Sur |
| PH22-0964 | - | *Fusarium* sp. (not *F. oxysporum* species complex) | Finca Verde, Tugbok, Davao City, Davao del Sur |
| PH22-0966 | - | *Fusarium* sp. (not *F. oxysporum* species complex) | Finca Verde, Tugbok, Davao City, Davao del Sur |
| PH22-0967 | - | *Fusarium* sp. (not *F. oxysporum* species complex) | Finca Verde, Tugbok, Davao City, Davao del Sur |
| PH22-0968 | - | *Fusarium* sp. (*F. oxysporum* species complex) | Finca Verde, Tugbok, Davao City, Davao del Sur |
| PH22-0969 | - | *Fusarium* sp. (not *F. oxysporum* species complex) | Finca Verde, Tugbok, Davao City, Davao del Sur |
| PH22-0970 | - | *Fusarium* sp. (not *F. oxysporum* species complex) | Finca Verde, Tugbok, Davao City, Davao del Sur |
| PH22-0971 | - | *Fusarium* sp. (not *F. oxysporum* species complex) | Finca Verde, Tugbok, Davao City, Davao del Sur |
| PH22-0972 | - | *Fusarium* sp. (*F. oxysporum* species complex) | Finca Verde, Tugbok, Davao City, Davao del Sur |
| PH22-0973 | - | *Fusarium* sp. (not *F. oxysporum* species complex) | Finca Verde, Tugbok, Davao City, Davao del Sur |
| PH22-0974 | - | *Fusarium* sp. (*F. oxysporum* species complex) | Finca Verde, Tugbok, Davao City, Davao del Sur |
| PH22-0979 | - | *Fusarium* sp. (not *F. oxysporum* species complex) | Finca Verde, Tugbok, Davao City, Davao del Sur |
| PH22-0981 | - | aff. *F. fabacearum* | Finca Verde, Tugbok, Davao City, Davao del Sur |
| PH22-0983 | - | *Fusarium* sp. (not *F. oxysporum* species complex) | Finca Verde, Tugbok, Davao City, Davao del Sur |
| PH22-0984 | - | *Fusarium* sp. (not *F. oxysporum* species complex) | Finca Verde, Tugbok, Davao City, Davao del Sur |
| PH22-0985 | + | Foc TR4 (cf. *F. odoratissimum*) | Finca Verde, Tugbok, Davao City, Davao del Sur |
| PH22-0986 | + | Foc TR4 (cf. *F. odoratissimum*) | Finca Verde, Tugbok, Davao City, Davao del Sur |
| PH22-0987 | + | Foc TR4 (cf. *F. odoratissimum*) | Finca Verde, Tugbok, Davao City, Davao del Sur |
| PH22-0987 | - | *Fusarium* sp. (not *F. oxysporum* species complex) | Finca Verde, Tugbok, Davao City, Davao del Sur |
| PH22-0988 | - | *Fusarium* sp. (not *F. oxysporum* species complex) | Finca Verde, Tugbok, Davao City, Davao del Sur |
| PH22-0995 | - | *Fusarium* sp. (not *F. oxysporum* species complex) | Finca Verde, Tugbok, Davao City, Davao del Sur |
| PH22-0998 | - | cf. *F. elaeidis* | Finca Verde, Tugbok, Davao City, Davao del Sur |
| PH22-1003 | - | *Fusarium* sp. (not *F. oxysporum* species complex) | Finca Verde, Tugbok, Davao City, Davao del Sur |
| PH22-1005 | - | *Fusarium* sp. (not *F. oxysporum* species complex) | Finca Verde, Tugbok, Davao City, Davao del Sur |
| PH22-1006 | + | *Fusarium* sp. (not *F. oxysporum* species complex) | Finca Verde, Tugbok, Davao City, Davao del Sur |

(*Continued*)

**Table 3.** (Continued)

| Isolates | Detection | Phylogenetic position | Location |
|---|---|---|---|
| PH22-1011 | + | *Fusarium* sp. (not *F. oxysporum* species complex) | Finca Verde, Tugbok, Davao City, Davao del Sur |
| PH22-1012 | - | cf. *F. triseptatum* | Finca Verde, Tugbok, Davao City, Davao del Sur |
| PH22-1013 | - | cf. *F. elaeidis* | Finca Verde, Tugbok, Davao City, Davao del Sur |
| PH22-1023 | - | *Fusarium* sp. (not *F. oxysporum* species complex) | Finca Verde, Tugbok, Davao City, Davao del Sur |
| PH22-1024 | - | *Fusarium* sp. (not *F. oxysporum* species complex) | Finca Verde, Tugbok, Davao City, Davao del Sur |
| PH22-1026 | - | *Fusarium* sp. (not *F. oxysporum* species complex) | Finca Verde, Tugbok, Davao City, Davao del Sur |
| PH22-1069 | + | *Fusarium* sp. (*F. oxysporum* species complex) | Tamayong, Davao City, Davao del Sur |
| PH22-1165 | + | cf. *F. triseptatum* | Biao, Guianga, Tugbok, Davao City, Davao del Sur |
| PH22-1166 | - | cf. *F. elaeidis* | Biao, Guianga, Tugbok, Davao City, Davao del Sur |
| PH22-1168 | - | cf. *F. triseptatum* | Biao, Guianga, Tugbok, Davao City, Davao del Sur |
| PH22-1169 | + | *Fusarium* sp. (not *F. oxysporum* species complex) | Biao, Guianga, Tugbok, Davao City, Davao del Sur |
| PH22-1203 | + | Foc TR4 (cf. *F. odoratissimum*) | Biao, Guianga, Tugbok, Davao City, Davao del Sur |
| PH22-1204 | - | *Fusarium* sp. (not *F. oxysporum* species complex) | Biao, Guianga, Tugbok, Davao City, Davao del Sur |
| PH22-1205 | - | *Fusarium* sp. (not *F. oxysporum* species complex) | Biao, Guianga, Tugbok, Davao City, Davao del Sur |
| PH22-1228 | + | Foc TR4 (cf. *F. odoratissimum*) | Biao, Guianga, Tugbok, Davao City, Davao del Sur |
| PH22-1229 | - | *Fusarium* sp. (not *F. oxysporum* species complex) | Biao, Guianga, Tugbok, Davao City, Davao del Sur |
| PH22-1230 | - | *Fusarium* sp. (not *F. oxysporum* species complex) | Biao, Guianga, Tugbok, Davao City, Davao del Sur |
| PH22-1234 | - | *Fusarium* sp. (not *F. oxysporum* species complex) | Biao, Guianga, Tugbok, Davao City, Davao del Sur |
| PH22-1235 | - | *Fusarium* sp. (not *F. oxysporum* species complex) | Biao, Guianga, Tugbok, Davao City, Davao del Sur |
| PH22-1237 | - | *Fusarium* sp. (not *F. oxysporum* species complex) | Biao, Guianga, Tugbok, Davao City, Davao del Sur |
| PH22-1238 | - | *Fusarium* sp. (not *F. oxysporum* species complex) | Biao, Guianga, Tugbok, Davao City, Davao del Sur |
| PH22-1241 | - | *Fusarium* sp. (not *F. oxysporum* species complex) | Biao, Guianga, Tugbok, Davao City, Davao del Sur |
| PH22-1244 | - | *Fusarium* sp. (not *F. oxysporum* species complex) | Biao, Guianga, Tugbok, Davao City, Davao del Sur |
| PH22-1249 | - | *Fusarium* sp. (not *F. oxysporum* species complex) | Biao, Guianga, Tugbok, Davao City, Davao del Sur |
| PH22-1250 | - | *Fusarium* sp. (not *F. oxysporum* species complex) | Biao, Guianga, Tugbok, Davao City, Davao del Sur |
| PH22-1251 | - | *Fusarium* sp. (not *F. oxysporum* species complex) | Biao, Guianga, Tugbok, Davao City, Davao del Sur |
| PH22-1252 | - | *Fusarium* sp. (not *F. oxysporum* species complex) | Biao, Guianga, Tugbok, Davao City, Davao del Sur |
| PH22-1256 | - | *Fusarium* sp. (not *F. oxysporum* species complex) | Biao, Guianga, Tugbok, Davao City, Davao del Sur |
| PH22-1257 | - | *Fusarium* sp. (not *F. oxysporum* species complex) | Biao, Guianga, Tugbok, Davao City, Davao del Sur |
| PH22-1260 | - | *Fusarium* sp. (not *F. oxysporum* species complex) | Biao, Guianga, Tugbok, Davao City, Davao del Sur |
| PH22-1265 | - | *Fusarium* sp. (not *F. oxysporum* species complex) | Biao, Guianga, Tugbok, Davao City, Davao del Sur |
| PH22-1267 | - | *Fusarium* sp. (not *F. oxysporum* species complex) | Biao, Guianga, Tugbok, Davao City, Davao del Sur |
| PH22-1268 | - | *Fusarium* sp. (not *F. oxysporum* species complex) | Biao, Guianga, Tugbok, Davao City, Davao del Sur |
| PH22-1270 | - | *Fusarium* sp. (not *F. oxysporum* species complex) | Biao, Guianga, Tugbok, Davao City, Davao del Sur |
| PH22-1271 | - | *Fusarium* sp. (not *F. oxysporum* species complex) | Biao, Guianga, Tugbok, Davao City, Davao del Sur |
| PH22-1272 | - | *Fusarium* sp. (not *F. oxysporum* species complex) | Biao, Guianga, Tugbok, Davao City, Davao del Sur |
| PH22-1273 | - | *Fusarium* sp. (not *F. oxysporum* species complex) | Biao, Guianga, Tugbok, Davao City, Davao del Sur |
| PH22-1274 | - | *Fusarium* sp. (not *F. oxysporum* species complex) | Biao, Guianga, Tugbok, Davao City, Davao del Sur |
| PH22-1276 | - | *Fusarium* sp. (not *F. oxysporum* species complex) | Biao, Guianga, Tugbok, Davao City, Davao del Sur |
| PH22-1277 | - | *Fusarium* sp. (not *F. oxysporum* species complex) | Biao, Guianga, Tugbok, Davao City, Davao del Sur |

'+' indicates that the strain was detected with the primers.

'-' indicated that the strain was not detected with the primers.

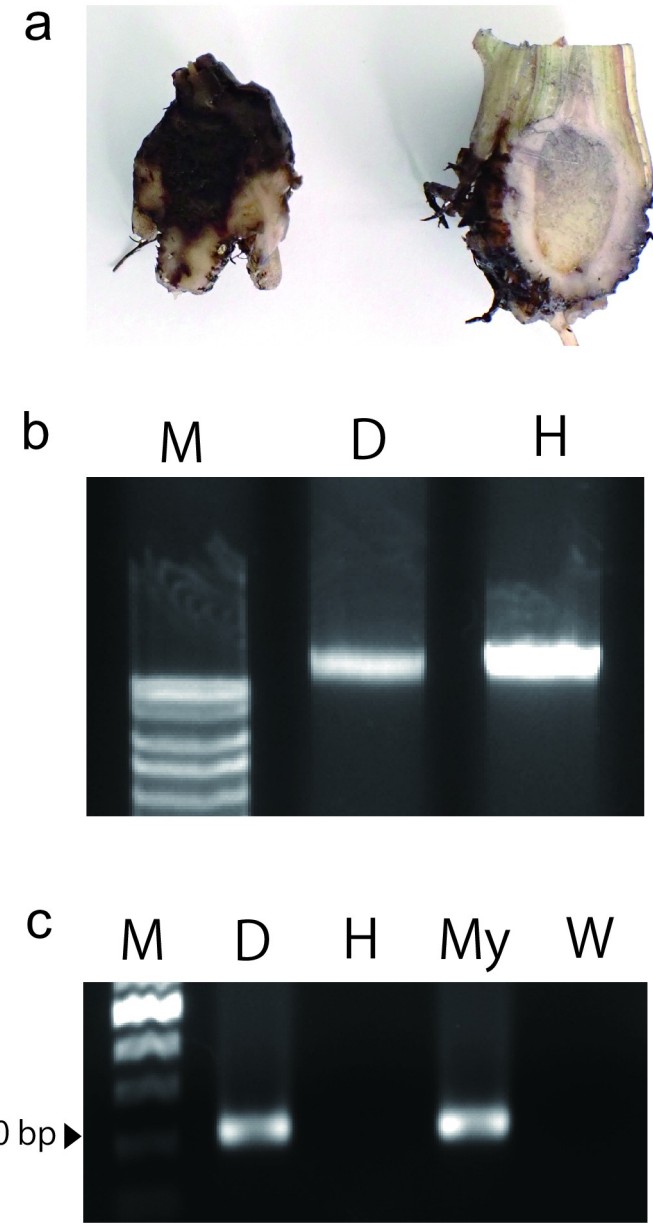

**Fig 5. Detection in plant tissues by PCR amplification. a** Diseased (left) and healthy (right) corms of dwarf cavendish banana with and without inoculation of Foc TR4 (2718M). **b** Extracted DNA from healthy and diseased tissues. Lane M: 1 kb DNA ladder; Lane D: genome DNA from diseased tissue; Lane H: genome DNA from healthy tissue. **c** Detection of PCR amplicons using DNA templates from healthy and diseased tissues and mycelia of Foc TR4 (2718M). Lane M: 100 bp DNA ladder. Lane D: PCR amplicon from diseased tissue. Lane H: PCR amplicon from healthy tissue. Lane My: PCR amplicon from mycelia. Lane W: Negative control with ddH$_2$O.

## Discussion

Fusarium wilt caused by Foc TR4 seriously threatens the productivity of the global banana industry. Rapid detection is important to prevent the spread of pathogens in the absence of resistant varieties. In this study, we designed more accurate and specific PCR primers for Foc TR4 than the previously developed primers. This study explored the genes specifically

possessed by Foc TR4 from the genome data of 93 *F. oxysporum* strains and identified gene 13712 on which to base the primer set. Primers were validated by PCR using the Foc TR4 strain and other formae speciales (Table 1, S3 Fig). Compared to the primers developed by Dita et al. [7], Carvalhais et al. [10], Li et al. [8], and Li et al. [9], which did not detect Foc TR4 or detected other formae speciales, the primers designed in this study detected only Foc TR4 (Table 1, S3 Fig). These results indicate that the PCR primer set 13712F and 13712R is more accurate than previously reported primers.

We verified that the primers designed in this study effectively detected Foc TR4 among the 86 isolates of *Fusarium* spp. collected from banana fields in the Philippines based on the monophyletic classification of the race [6]. The PCR detection results identified seven strains that were monophyletic with Foc TR4, with no strains remaining undetected, at least as false negatives. In contrast, four of the 86 strains were detected as false positives. This may be because the target gene was horizontally transferred or this region, which was inherited from a common ancestor, remained intact in strains other than Foc TR4. However, based on our comparison of primer specificities by PCR trials including other formae speciales, the primer set 13712F and 13712R offered better detection (Table 1; S1 Fig), although not complete, because some isolates (4 out of 79 non-Foc TR4 strains) that are from fields and not Foc TR4 were detected. Furthermore, our primers successfully detected Foc TR4 in diseased plant tissue (Fig 5) and are expected to reduce the time and effort required for diagnosis.

As more regions and high-quality genome data, such as at the chromosomal level, become available for a wider range of lineages, so does the availability of appropriate sequences for designing Foc TR4-specific PCR primers. As with the genome data, further investigation into relevant strains is required to confirm the reliability of the designed primers. In this study, primer specificity was evaluated on 18 formae speciales, including Foc TR4 and other formae speciales strains. However, >100 *F. oxysporum* formae speciales have been reported up to August 2018 [26]. Since it is not realistic to evaluate primer specificity for all of them, further evaluation will be needed across various future studies on banana fusarium wilt.

Given the rapid evolution of plant pathogens, monitoring should be performed not only for Foc TR4 but also other potentially harmful pathogens. Therefore, pathogen detection using specific PCR primers should be performed periodically according to up-to-date classification systems and diagnosed based on Koch's principles.

## Supporting information

**S1 Fig. Morophology of 2718M (*Fusarium oxysporum* f. sp. *cubense*).**
(PDF)

**S2 Fig. The result of pathogenicity test of *F. oxysporum* f. sp. *cubence* tropical race 4 (2718M) to bananas (cv. dwarf Cavendish).**
(PDF)

**S3 Fig. PCR detection of Foc TR4 using specific primers sets 13712F/13712R (this study), Foc TR4F/Foc TR4R [7], W2987F/W2987R [10], SIX1a-266-F/SIX1a-266-R [9], and VCG01213 16F/VCG01213 16R [8].** Lane M, marker; lane 1, Foc TR4; lane 2, Foc race 1; lane 3, Foc race 2; lane 4, MAFF103008; lane 5, MAFF235154; lane 6, MAFF240805; lane 7, MAFF237022; lane 8, MAFF243476; lane 9, MAFF727508; lane 10, MAFF306716; lane 11, MAFF744004; lane 12, MAFF305544; lane 13, MAFF240804; lane 14, MAFF243255; lane 15, MAAFF241054; lane 16, MAFF306313; lane 17, MAFF305937; lane 18, MAFF103036; lane 19, MAFF235105; lane 20, MAFF744088; lane 21, MAFF103054; lane 22, MAFF305606; lane 23, MAFF240327; lane 24, MAFF103059; lane 25, MAFF103051; lane 26, MAFF240102; lane 27,

MAFF305543; lane 28, MAFF305608; lane C, ddH2O.
(PDF)

**S4 Fig. PCR analysis of *Fusarium* spp. using primer set 13712F/13712R.** Isolate names are shown in each lane. The first lane of each panel (M) shows the marker sizes using the Excel-Band™ 100 bp DNA ladder (SMOBIO). The upper and lower panels show the amplification results for the ITS and 13712 regions, respectively. Isolates with red characters indicate amplification using primer set 13712F/13712R. Isolates with red dots indicate that isolates were identified as Foc TR4.
(PDF)

**S1 Table. GenBank accession numbers for genome data.**
(XLSX)

**S2 Table. List of FocTR4 specific primers used in this study.**
(XLSX)

**S1 Data. Sequence of an amplicon obtained from 2718M by PCR using primer set 13712F/13712R.**
(TXT)

**S1 Dataset. Sequence data sets for ITS.**
(TXT)

**S2 Dataset. Sequence data sets for *cmdA*, *rpb2*, *tef1*, and *tub2*.**
(ZIP)

## Acknowledgments

We thank Dr. Reynaldo Valle, Mr. Yoshiki Takata, Ms. Yui Harada, and Ms. Yuriko Sakurai (Tamagawa University BacaDM Project) for supporting our study. We are grateful to Ms. Celynne Ocampo-Padilla for giving us advice on writing this paper.

## Author Contributions

**Conceptualization:** Shunsuke Nozawa, Kyoko Watanabe.

**Formal analysis:** Shunsuke Nozawa.

**Funding acquisition:** Kyoko Watanabe.

**Investigation:** Shunsuke Nozawa, Dan Charlie Joy Pangilinan, G. Alvindia Dionisio, Kyoko Watanabe.

**Methodology:** Shunsuke Nozawa.

**Project administration:** Kyoko Watanabe.

**Software:** Shunsuke Nozawa.

**Supervision:** Kyoko Watanabe.

**Validation:** Shunsuke Nozawa.

**Visualization:** Shunsuke Nozawa.

**Writing – original draft:** Shunsuke Nozawa.

**Writing – review & editing:** Kyoko Watanabe.

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
