## [Decision Letter · Decision Letter 0]

10 Sep 2024

PONE-D-24-28376Specific PCR primer designed from genome data for rapid detection of Fusarium oxysporum f. sp. cubense tropical race 4 in the Cavendish bananaPLOS ONE

Dear Dr. Watanabe,

Thank you for submitting your manuscript to PLOS ONE. After careful consideration, we feel that it has merit but does not fully meet PLOS ONE’s publication criteria as it currently stands. Therefore, we invite you to submit a revised version of the manuscript that addresses the points raised during the review process.

We look forward to receiving your revised manuscript.

Kind regards,

Ravinder Kumar, Ph.D.

Academic Editor

PLOS ONE

Journal Requirements:

1. When submitting your revision, we need you to address these additional requirements. Please ensure that your manuscript meets PLOS ONE's style requirements, including those for file naming. The PLOS ONE style templates can be found at https://journals.plos.org/plosone/s/file?id=wjVg/PLOSOne_formatting_sample_main_body.pdf and https://journals.plos.org/plosone/s/file?id=ba62/PLOSOne_formatting_sample_title_authors_affiliations.pdf 2. Please expand the acronym “JICA JST” (as indicated in your financial disclosure) so that it states the name of your funders in full. This information should be included in your cover letter; we will change the online submission form on your behalf. 3. Thank you for stating the following financial disclosure: "JICA JST SATREPS Grant Number JPMJSA2007" Please state what role the funders took in the study.  If the funders had no role, please state: "The funders had no role in study design, data collection and analysis, decision to publish, or preparation of the manuscript." If this statement is not correct you must amend it as needed. Please include this amended Role of Funder statement in your cover letter; we will change the online submission form on your behalf.

Reviewers' comments:

Reviewer's Responses to Questions

**Comments to the Author**

1. Is the manuscript technically sound, and do the data support the conclusions?

Reviewer #1: Yes

Reviewer #2: Yes

Reviewer #3: Yes

2. Has the statistical analysis been performed appropriately and rigorously? 

Reviewer #1: Yes

Reviewer #2: Yes

Reviewer #3: Yes

3. Have the authors made all data underlying the findings in their manuscript fully available?

Reviewer #1: Yes

Reviewer #2: Yes

Reviewer #3: Yes

4. Is the manuscript presented in an intelligible fashion and written in standard English?

Reviewer #1: No

Reviewer #2: Yes

Reviewer #3: Yes

5. Review Comments to the Author

Reviewer #1: The research entitled "Specific PCR primer designed from genome data for rapid detection of Fusarium oxysporum f. sp. cubense tropical race 4 in Cavendish banana" is a good research, the authors did an excellent job, however, I recommend some changes to improve the formatting of the manuscript.

1. Expand the introduction, demonstrating the history of Fusarium wilt in bananas, for example, write a history since the discovery of banana wilt in Central America, going through the morphology, phylogenetic analysis (for example, how many lineages are there within Fusarium oxysporum f.sp. cubense). I found the authors' approach interesting, even designing the primers makes it clear that race 4 is a species called Fusarium odoratissimum.

2. I recommend including in the material and methods photos of plants with symptoms of plants with wilt, or were the isolates obtained from a collection? 3. The authors do not mention the other isolates identified in the phylogeny, for example isolate PH 22-981 grouped with Fusarium fabacearum, the isolates grouped with Fusarium elaeidis, however, these results are not valued in the discussion.

4.The authors should value the phylogeny data, at no point do they mention the grouping of the isolates studied, please look at the phylogenetic tree and describe the data obtained.

5. What was the morphology of the isolates? Why not include a figure with the fungus cultures?

6. In the discussion, the authors mention this information: "However, based on our comparison of primer specificities, the primer set 13712F and 13712R may offer better detection, although they are not as perfect as the other primers because some isolates (4 out of 79 non-Foc TR4 strains) that are from fields and not Foc TR4 were detected. Therefore, our primer successfully detected Foc TR4 in diseased plant tissue (Fig 5) and are expected to reduce the time and effort required for diagnosis". This worries me, specific primers should be perfect and the authors mention that they are not so perfect, they should justify this sentence better.

7. Although the research is important, I see that the manuscript needs to be improved.... I noticed that the main focus was only the design of the primers, but in addition, the authors obtained other results and this should be discussed.

Reviewer #2: Fusarium oxysporum f. sp. cubense (Foc) tropical race 4 (TR4) causes severely banana wilt. The rapid diagnosis is necessary to monitor the disease outbreaks. Current detection methods, even including race specific primer pairs, have the defects on time consuming or accuracy. The author Kyoko Watanabe developed a new primer pair based on one race specific gene from comparing genome analysis. Application of the new primer pair increased the accuracy and efficiency to investigate the TR4 distribution. The findings has the potential epidemiological significance to increase the level of monitoring TR4 population. My concerns are as follows.

1. Line 130: It’s better to provide a list of the 86 isolates with locations as well as the positive and negative reference strains that would be used during the PCR screening.

2. Line 170: Please make sure Fig.1 legend for Fig. 1 or Fig. 2 (Line 187), and vice versa. Fig.2 just shows the locations of two primer fragments.

3. Line 181: Even gene 13712 is specific to TR4, aliment of the sequence through NCBI in case of false positive result. No more illustration about the sequence of unique gene 13712 and how to design the sequence. Is there any another better one?

4. Line 242: The legend should mention each line referring to and whether the positive, negative, empty control went together with.

Reviewer #3: This study examined the genome data of 93 Fusarium oxysporum strains to identify genes unique to F. oxysporum f. sp. cubense Tropical Race 4 (Foc TR4), ultimately selecting Foc TR4-specific loci as the basis for developing a new PCR primer set. The primers were validated using Foc TR4 and other related strains. Compared to primers designed in previous studies, which either failed to detect Foc TR4 or detected non-TR4 strains, the newly designed primers specifically detected Foc TR4.

When tested on 86 isolates from banana fields in the Philippines, the new primer set accurately identified seven strains as Foc TR4, with no false negatives. However, four non-TR4 strains were detected as false positives, potentially due to horizontal gene transfer or shared ancestral genetic regions. Despite these minor false positives, the 13712F and 13712R primers proved more accurate than previously reported primers. The study concludes that the new primers will improve the speed and accuracy of diagnosing Foc TR4 in banana plants.

In summary, the authors of the manuscript under review developed a new PCR primer set based on a pathogenicity-related gene, specifically for detecting Fusarium oxysporum f. sp. cubense Tropical Race 4 (Foc TR4). Compared to previous primers, these showed greater specificity in identifying Foc TR4, with no false negatives and minimal false positives when tested on 86 isolates from banana fields. While not perfect, the new primers offer improved accuracy for diagnosing Foc TR4 and are expected to streamline the diagnostic process in banana fields.

The paper reports on an experiment conducted to a high technical standard and reports new results that should prove beneficial to the broader scientific community. Therefore, I recommend this manuscript for publication.

6. PLOS authors have the option to publish the peer review history of their article (what does this mean?). If published, this will include your full peer review and any attached files.

Reviewer #1: No

Reviewer #2: No

Reviewer #3: **Yes: **Joseph Flaherty

---

## [Author Response · Author response to Decision Letter 0]

13 Oct 2024

RESPONSE TO REVIEWERS

Thank you for considering our manuscript (PONE-D-24-28376), now we revised sentences according to the reviewer’s advice. We particularly thank the reviewer for his/her incisive comment, which has helped us to make significant improvements to the scientific aspects of our study. Our responses to the comments are as follows.

RESPONSE TO REVIEWE#1

Comments from Reviewer #1:

1. Expand the introduction, demonstrating the history of Fusarium wilt in bananas, for example, write a history since the discovery of banana wilt in Central America, going through the morphology, phylogenetic analysis (for example, how many lineages are there within Fusarium oxysporum f.sp. cubense). I found the authors' approach interesting, even designing the primers makes it clear that race 4 is a species called Fusarium odoratissimum.

Response

Thank you so much for giving us advice for the introduction. I have followed your advice and made the following corrections.

Line 34-38

“However, banana production has been threatened by banana wilt disease caused by Fusarium oxysporum f. sp. cubense (Foc), a soil-borne pathogen. In the 20th century, the disease caused by Foc race 1 eradicated thousands of hectares of the Gros Michel banana in Central America. In response to this, the resistant variety, Cavendish banana was discovered and this variety became the dominant commercial banana worldwide.”

Line 49-55

“A recent study showed that although it is difficult to find differences in morphology from other races, Foc TR4 is independent of other lineages including Foc race 1 and Foc subtropical race 4 (STR4), based on phylogenetic analysis using four loci: calmodulin (cmdA), RNA polymerase second largest subunit (rpb2), translation elongation factor 1-alpha (tef1), and β-tubulin 2 (tub2) regions [6]. Therefore, Foc TR4 can be distinguished from other races by phylogenetic analysis.”

Comments from Reviewer #1:

2. I recommend including in the material and methods photos of plants with symptoms of plants with wilt, or were the isolates obtained from a collection? 3. The authors do not mention the other isolates identified in the phylogeny, for example isolate PH 22-981 grouped with Fusarium fabacearum, the isolates grouped with Fusarium elaeidis, however, these results are not valued in the discussion.

Response

We appreciate your suggestion that symptoms be shown. However, we do not have a picture of the original symptoms. So, instead of that, we added a result of the pathogenicity test against the Cavendish banana as supplemental data (S2 Fig) to show that 2718M strain is Foc TR4. The text below was added accordingly.

Line 111-112

“Foc TR4 isolates (2718M; S1 Fig) that confirmed its pathogenicity against the Cavendish variety and (S2 Fig),…”

Comments from Reviewer #1:

4.The authors should value the phylogeny data, at no point do they mention the grouping of the isolates studied, please look at the phylogenetic tree and describe the data obtained.

Response

Thank you so much for the valuable advice on the phylogeny of isolates. We agreed with your suggestion and added descriptions of the phylogenetic positions of our isolates in addition to that of Foc TR4 as below.

Line 226-233

“Other isolates were clustered with F. elaeidis (5 isolates; ML/NJ/MP = 73%/71%/84%), F. fabacearum (one isolate; ML/NJ/MP = -/-/78%), F. trachichlamydosporum (two isolates; ML/NJ/MP = 98%/98%/99%), and F. triseptatum (two isolates; ML/NJ/MP = 99%/99%/99%). Four isolates are independent of known species and closely related to F. glycines. One isolate, PH22-1069, of those, and three isolates, PH22-1006, PH22-1011, and PH22-1023, from other species complexes (not determined; Fig. 3) were detected using this primer set were not Foc TR4 (Figs 3 and 4, S4 Fig).”

Comments from Reviewer #1:

5. What was the morphology of the isolates? Why not include a figure with the fungus cultures?

Response

Thank you so much for your suggestion about the morphologies of the fungi. For 2718M, which was used for verification, we have included a photo of the isolate as supplemental data for reference (S1 Fig; Line 111). However, we did not include other isolates. This is because FocTR4 has been reported to be monophyletic in phylogenetic analysis, and it is sufficient to show the phylogenetic relationship to validate the primers.

Comments from Reviewer #1:

6. In the discussion, the authors mention this information: "However, based on our comparison of primer specificities, the primer set 13712F and 13712R may offer better detection, although they are not as perfect as the other primers because some isolates (4 out of 79 non-Foc TR4 strains) that are from fields and not Foc TR4 were detected. Therefore, our primer successfully detected Foc TR4 in diseased plant tissue (Fig 5) and are expected to reduce the time and effort required for diagnosis". This worries me, specific primers should be perfect and the authors mention that they are not so perfect, they should justify this sentence better.

Response

Thank you very much for the advice on the justification for the utilization of the developed primer set. We agree with your suggestion. The primers we developed would detect some other strains. However, they have proven to give better results when compared to primers developed previously. Therefore, we have revised the sentences as follows.

Line 287-291

“However, based on our comparison of primer specificities by PCR trials including other formae speciales, the primer set 13712F and 13712R offered better detection (Table 1; S1 Fig), although not complete because some isolates (4 out of 79 non-Foc TR4 strains) that are from fields and not Foc TR4 were detected. Furthermore, our primers successfully detected Foc TR4 in diseased plant tissue (Fig 5) and are expected to reduce the time and effort required for diagnosis.”

Comments from Reviewer #1:

7. Although the research is important, I see that the manuscript needs to be improved.... I noticed that the main focus was only the design of the primers, but in addition, the authors obtained other results and this should be discussed.

Response

We appreciate your suggestion. We agree with your advice. Without changing the main trend of validating the usefulness of the developed primers, we added to the results the phylogenetic relationships of the isolates obtained by phylogenetic analysis.

As described in the responses to suggestions 3 and 4, we added sentences as below.

Line 226-233

“Other isolates were clustered with F. elaeidis (5 isolates; ML/NJ/MP = 73%/71%/84%), F. fabacearum (one isolate; ML/NJ/MP = -/-/78%), F. trachichlamydosporum (two isolates; ML/NJ/MP = 98%/98%/99%), and F. triseptatum (two isolates; ML/NJ/MP = 99%/99%/99%). Four isolates are independent of known species and closely related to F. glycines. One isolate, PH22-1069, of those, and three isolates, PH22-1006, PH22-1011, and PH22-1023, from other species complexes (not determined; Fig. 3) were detected using this primer set were not Foc TR4 (Figs 3 and 4, S4 Fig).” 

RESPONSE TO REVIEWE#2

Comments from Reviewer #2:

1. Line 130: It’s better to provide a list of the 86 isolates with locations as well as the positive and negative reference strains that would be used during the PCR screening.

Response

Line 251

We appreciate your suggestion. We agree with that we add the list of 86 isolates. We added a table (Table 3) which contains the result of detection from collected isolates, and their locations, and their phylogenetic positions.

Comments from Reviewer #2:

2. Line 170: Please make sure Fig.1 legend for Fig. 1 or Fig. 2 (Line 187), and vice versa. Fig.2 just shows the locations of two primer fragments.

Response

Thank you so much for your correction. We replaced Fig. 1 and Fig. 2 in the proper place.

Line 177-180 Fig 1

Line 198-199 Fig 2

Comments from Reviewer #2:

3. Line 181: Even gene 13712 is specific to TR4, aliment of the sequence through NCBI in case of false positive result. No more illustration about the sequence of unique gene 13712 and how to design the sequence. Is there any another better one?

Response

Thank you so much for your suggestion about the sequence of the target of the specific primers. 

Line 192

We added the sequence data of the target region as supplemental data (S1 Data). 

Line 193-195

We added the following sentence as the information on the primer set's design. “For a forward primer 13712F, GC content is 40%, and the annealing temperature is 53.2℃. For a reverse primer13712R, GC content is 26.9%, and the annealing temperature is 54.6℃.”

Comments from Reviewer #2:

4. Line 242: The legend should mention each line referring to and whether the positive, negative, empty control went together with.

Response

We appreciate your correction. We have reflected on this comment and we changed the explanation of Figure 5 about the positive and negative control as follows.

Line 260-265

Fig 5. Detection in plant tissues by PCR amplification. a Diseased (left) and healthy (right) corms of dwarf cavendish banana with and without inoculation of Foc TR4 (2718M). b Extracted DNA from healthy and diseased tissues. Lane M: 1 kb DNA ladder; Lane D: genome DNA from diseased tissue; Lane H: genome DNA from healthy tissue. c Detection of PCR amplicons using DNA templates from healthy and diseased tissues and mycelia of Foc TR4 (2718M). Lane M: 100 bp DNA ladder. Lane D: PCR amplicon from diseased tissue. Lane H: PCR amplicon from healthy tissue. Lane My: PCR amplicon from mycelia. Lane W: Negative control with ddH2O.

RESPONSE TO REVIEWE#3

Response to Reviewer #3

Thank you very much for recommending our paper for publication. We have revised the content of the paper in accordance with the advice of reviewers 1 and 2.

---

## [Decision Letter · Decision Letter 1]

23 Oct 2024

Specific PCR primer designed from genome data for rapid detection of Fusarium oxysporum f. sp. cubense tropical race 4 in the Cavendish banana

PONE-D-24-28376R1

Dear Dr. Watanabe,

We’re pleased to inform you that your manuscript has been judged scientifically suitable for publication and will be formally accepted for publication once it meets all outstanding technical requirements.

Kind regards,

Ravinder Kumar, Ph.D.

Academic Editor

PLOS ONE

**Comments to the Author**

1. If the authors have adequately addressed your comments raised in a previous round of review and you feel that this manuscript is now acceptable for publication, you may indicate that here to bypass the “Comments to the Author” section, enter your conflict of interest statement in the “Confidential to Editor” section, and submit your "Accept" recommendation.

Reviewer #1: All comments have been addressed

Reviewer #3: All comments have been addressed

2. Is the manuscript technically sound, and do the data support the conclusions?

Reviewer #1: Yes

Reviewer #3: Yes

3. Has the statistical analysis been performed appropriately and rigorously? 

Reviewer #1: Yes

Reviewer #3: Yes

4. Have the authors made all data underlying the findings in their manuscript fully available?

Reviewer #1: Yes

Reviewer #3: Yes

5. Is the manuscript presented in an intelligible fashion and written in standard English?

Reviewer #1: Yes

Reviewer #3: Yes

6. Review Comments to the Author

Reviewer #1: The authors responded to all the questions and suggestions requested, and therefore I recommend that the manuscript be accepted for publication. Obviously, the primer for the detection of TR4 is an important diagnostic methodology.

Reviewer #3: The authors appear to have carefully considered the many helpful comments expressed by the expert reviews upon preparing an improved manuscript suitable for publication.

7. PLOS authors have the option to publish the peer review history of their article (what does this mean?). If published, this will include your full peer review and any attached files.

Reviewer #1: No

Reviewer #3: No

---

## [Editor Report · Acceptance letter]

19 Nov 2024

PONE-D-24-28376R1 

PLOS ONE

Dear Dr. Watanabe, 

I'm pleased to inform you that your manuscript has been deemed suitable for publication in PLOS ONE. Congratulations! Your manuscript is now being handed over to our production team.

Kind regards, 

on behalf of

Dr. Ravinder Kumar 

Academic Editor

PLOS ONE